# Source reduction with a purpose: Mosquito ecology and community perspectives offer insights for improving household mosquito management in coastal Kenya

Jenna E. Forsyth[1]*, Francis M. Mutuku[2,3], Lydiah Kibe[4], Luti Mwashee[2,3], Joyce Bongo[3], Chika Egemba[5], Nicole M. Ardoin[6], A. Desiree LaBeaud[5]

1 Emmett Interdisciplinary Program in Environment and Resources, Stanford University, Stanford, California, United States of America, 2 Technical University of Mombasa, Mombasa, Kenya, 3 Vector-Borne Disease Unit, Msambweni County Referral Hospital, Msambweni, Kenya, 4 Centre for Geographic Medicine Research Coast, Kenya Medical Research Institute, Kilifi, Kenya, 5 Stanford University School of Medicine, Stanford, California, United States of America, 6 Emmett Interdisciplinary Program in Environment and Resources, Graduate School of Education, and Woods Institute for the Environment, Stanford University, Stanford, California, United States of America

* jforsyth@stanford.edu

## Abstract

Understanding mosquito breeding behavior as well as human perspectives and practices are crucial for designing interventions to control *Aedes aegypti* mosquito-borne diseases as these mosquitoes primarily breed in water-holding containers around people's homes. The objectives of this study were to identify productive mosquito breeding habitats in coastal Kenya and to understand household mosquito management behaviors and their behavioral determinants. The field team conducted entomological surveys in 444 households and semi-structured interviews with 35 female caregivers and 37 children in Kwale County, coastal Kenya, between May and December 2016. All potential mosquito habitats with or without water were located, abundances of mosquito immatures measured and their characteristics recorded. Interviews explored household mosquito management behaviors and their behavioral determinants. 2,452 container mosquito habitats were counted containing 1,077 larvae and 390 pupae, predominantly *Aedes* species. More than one-third of the positive containers were found outside houses in 1 of the 10 villages. Containers holding water with no intended purpose contained 55.2% of all immature mosquitoes. Containers filled with rainwater held 95.8% of all immature mosquitoes. Interviews indicated that households prioritize sleeping under bednets as a primary protection against mosquito-borne disease because of concern about night-time biting, malaria-transmitting *Anopheles* mosquitoes. Respondents had limited knowledge about the mosquito life cycle, especially with respect to day-time biting, container-breeding *Aedes* mosquitoes. Therefore, respondents did not prioritize source reduction. Most mosquitoes breed in containers that have no direct or immediate purpose ("no-purpose containers"). These containers may be left unattended for several days allowing rainwater to collect, and creating ideal conditions for mosquito breeding. An

**Data Availability Statement:** All relevant data are within the manuscript and its Supporting Information files.

**Funding:** ADL received the Bechtel Faculty Scholar award from Stanford's Maternal & Child Health Research Institute (no grant number). (https://med.stanford.edu/mchri.html) JEF received an award from Stanford's Center for African Studies (no grant number). (https://africanstudies.stanford.edu/) JEF received an award from the Emmett Interdisciplinary Program in Environment and Resources (no grant number). (https://pangea.stanford.edu/eiper) The funders had no role in study design, data collection and analysis, decision to publish, or preparation of the manuscript.

**Competing interests:** The authors have declared that no competing interests exist.

intervention that requires little effort and targets only the most productive containers could effectively reduce mosquito indices and, relatedly, mosquito-borne disease risk.

## Author summary

Because *Ae. aegypti* mosquitoes bite during the day, bednets are not protective. Moreover, *Ae. aegypti* mosquitoes are 'anthropophilic container breeders' primarily breeding in containers outside people's homes. Therefore, vector control efforts that reduce the abundance of containers and other potential mosquito breeding habitats should be prioritized. This research aimed to identify productive *Ae. aegypti* mosquito breeding habitats in coastal Kenya and to understand household mosquito management behaviors and their behavioral determinants. We found that more than half of all immature mosquitoes were in containers with no intended purpose that had unintentionally filled with rainwater. Residents had limited awareness of day-time biting, container-breeding, *Aedes* mosquitoes. Consequently, households prioritized sleeping under bednets as a primary protection against mosquito-borne disease. Our findings inform the design of vector control efforts; encouraging community trash clean-up events and targeting the reduction or re-use of unused containers.

## Introduction

Emerging arthropod-borne viruses (arboviruses), which are spread by the *Aedes aegypti* mosquito, pose a substantial threat to global public health [1]. Unlike the night-time biting *Anopheles* mosquito that transmits malaria, *Ae. aegypti* bites during the day and transmits multiple arboviruses, including dengue, chikungunya, Zika, and yellow fever viruses. Individuals with these diseases can range from being asymptomatic to suffering from life-threatening encephalitis and hemorrhage, or debilitating arthritis that can persist for years [2]. Arboviral disease outbreaks have been unpredictable and increasing in frequency over the past two decades [3]. Kenya and other African countries have experienced a number of outbreaks in the past 5 years, despite little attention to the issue from government and community organizations [4]. In this context, outbreaks are often underreported and infections misdiagnosed as malaria [5–7].

Since there is no antiviral therapy and bednets are not protective against the day-time biting *Ae. aegypti* mosquitoes, it is imperative to focus vector control efforts on reducing the number of available *Ae. aegypti* breeding sites (source reduction). Individuals in the community play a crucial role in control efforts because *Ae. aegypti* mosquitoes are 'anthropophilic container breeders' primarily breeding in outdoor water containers such as buckets located immediately outside people's homes [8]. In Kenya, adult *Ae. aegypti* mosquitoes have been found to bite during circumscribed times of the day [9].

Source reduction may include tasks like covering containers, discarding containers, or cleaning outdoor environments. These behaviors can be numerous, complex, and difficult to perform, let alone sustain, by household members [10, 11]. Despite these challenges, community-based interventions promoting source reduction have effectively reduced mosquito indices in other countries by engaging women, who are often involved with water-related activities like fetching and storage [12], and children, who may be willing to engage with new ideas, more flexible in taking up behaviors early in life, and acting as agents of change in

communities [13]. Studies in South America and Asia have demonstrated reductions in mosquito indices from women and children's involvement in comprehensive source reduction of all potential mosquito habitats as well as targeted source reduction of containers previously identified as having the highest mosquito larval and pupal densities [11, 12].

The objectives of this study conducted in ten villages in Kwale County, coastal Kenya, were to identify productive mosquito breeding habitats outside homes and explore household mosquito management behaviors and their behavioral determinants among female caregivers and children. Together, this information could be used to develop community-based source reduction interventions that aim to target the most productive container habitats.

## Methods

### Study sites

This study was conducted in coastal rural villages near the town of Msambweni in Kwale County, Kenya, located approximately 60 kilometers south of Mombasa and 50 kilometers north of the Kenya-Tanzania border (4˚28′0.0114″S, 39˚28′0.12″E).

The annual mean temperatures range from 23–34˚C with average relative humidity between 60–80%. Precipitation varies throughout the year: February is the driest month, with an average of 18 mm of rain, and May is the wettest with an average of 347 mm. The seasons are classified based on precipitation levels with the long dry season between January-March, the long rainy season between April-June, the short dry season between July-September, and the short rainy season between October-December. With low population densities of 460 people/km$^2$, central water systems transporting piped water to households are lacking. As a result, residents obtain water for domestic purposes from rainfall in the wet months and wells and boreholes in the dry months. Fishing and subsistence farming are the primary livelihoods among residents. Islam is the dominant religion. [8, 14]

### Entomological surveys

We aimed to conduct entomological surveys in 500 households to understand container productivity profiles. We conducted these surveys primarily during the short rainy season between September and December 2016. Fifty houses with children in grades 5 and 6 (approximately ages 11 to 16) were randomly selected from 10 different primary school rosters. Outside each house, all potential larval habitats in the outdoor domestic environment of every house were inspected for mosquito larvae and pupae. We excluded the indoor environment because a prior study conducted by our team in the same region indicated that indoor habitats accounted for only 5.2% of the positive containers and were therefore deemed a lower priority than outdoor containers [8]. The larval habitats were classified into different habitat types (as described by [8]. All pupae and a sample of larvae (3$^{rd}$ and 4$^{th}$ instars) from positive larval habitats were collected with the aid of pipettes and ladles [15], counted and recorded on field-data forms. Technicians from the Msambweni Hospital Vector-borne Disease Control Unit reared the larvae and pupae to adult mosquitoes for species identification. Rearing conditions were kept stable in the laboratory at an average temperature of 28˚C and relative humidity of 80%. Larvae and pupae were kept in 200 ml plastic cups and fed TetraMinbaby® fish food (Tetra Werke, Melle, Germany). Standard taxonomic keys were used to distinguish *Ae. aegypti* species [16]. Characteristics were reported for each observed habitat, including the presence or absence of water, habitat type, size, purpose, water source, and frequency of filling and emptying. Purpose was identified in consultation with the female head of household who was asked an open-ended question about how the family was currently using each container or if the container had no immediate purpose.

Entomology survey data were analyzed using descriptive statistics of the number of habitats and number of productive habitats according to their type, purpose, and water source. Standard entomological indices were also calculated including the container index (percent of water-holding containers with larvae or pupae), Breteau index (number of positive containers per 100 houses), and house index (percent of houses with positive containers).

## Semi-structured interviews

A pilot entomological survey among 100 of the 500 households was conducted between May and July 2016. From the results of this pilot survey, 40 households were selected for semi-structured in-depth interviews based on approximated risk levels for mosquito-borne disease. The 20 'highest risk' households were selected because they had the most total containers and had the most mosquito larvae or pupae in containers. Conversely, the 20 'lowest risk' households were selected because had the fewest number of containers and they had no mosquito larvae or pupae in containers. Research assistants collected demographic data for 40 female caregivers and conducted 35 semi-structured in-depth interviews with these women lasting approximately 45 minutes. In addition to female caregivers, research assistants conducted semi-structured interviews with 37 of the women's children in grades 5–6 (ages 11–16).

The overarching goals of the interviews were to explore household mosquito management behaviors and their behavioral determinants. Research assistants were trained to be neutral and probe in a consistent manner (see S1 Text, S2 Text, S3 Text and S4 Text for English and Swahili versions of the interview protocol). To the extent possible, research assistants asked open-ended questions such as, "What do you know about mosquitoes?" The research assistants showed participants a video of mosquito larvae and pupae to elicit discussion around respondents' understanding about the mosquito life cycle.

Interviews were conducted in Kiswahili or Kidigo, depending on participant preference, and audio-recorded. Research assistants then transcribed and translated the interviews into English. Two coders who did not conduct the interviews analyzed the transcripts for themes. They used an *a priori* (deductive) and emergent (inductive) coding processes guided by our interest in mosquito-borne disease risk perception, and motivation to engage in source reduction and other protective behaviors. They identified themes in the interviews and analyzed data by reporting frequency of mentioning those themes by respondents.

## Ethics statement

We obtained written informed consent and assent from all study participants. The study protocol was reviewed and approved by the ethical review committee at the Kenyatta National Hospital/University of Nairobi (protocol # 241/03/2016) and the Institutional Review Board (IRB) of Stanford University (protocol #35504).

## Results

### Entomological surveys

A total of 2,452 container mosquito habitats were identified outside 444 houses across 10 villages. Among the 1,786 containers filled with water, 34 were positive (container index: 1.9%; Breteau index: 7.7). A total of 436 early instars, 641 late instars, and 390 pupae were identified. 82% were *Aedes aegypti*, and 18% were *Culex* species.

Positive containers were found outside 24 houses (house index: 5.4%), located in 5 of the 10 villages. More than one-third of the positive containers were found outside houses in 1 of the 10 villages. Among the 24 houses with positive containers, the average number of water-

holding containers was 6.2 (±3.1 S.D.), nearly 3 times higher than the average for all houses: 2.2 (±1.8 S.D.). Water-holding containers varied in size from small domestic containers and bottles (<5L) to large drums and tanks (>25L), though most containers were buckets and jerrycans (10-25L) (Fig 1).

More than half of all immature mosquitoes (55.2%) were found in tires, buckets, and small domestic containers with no immediate purpose. Buckets for laundry were the next most productive, containing 37.4% of immature mosquitoes. Although tires accounted for less than 1% of all containers, they contained 28.0% of immature mosquitoes. Containers used for all other purposes were minimally productive even though they were more abundant. The majority of positive containers, and the most highly productive containers, held rainwater. These accounted for 95.8% of immature mosquitoes. (Table 1, Table 2, Table 3, Table 4, S1 Table and S2 Table)

## Semi-structured interviews

The average age of the female caregivers was 37.2 (±9.2) with an average of 4.7 (±3.6) years of education. Most women engaged in farming or small business-related activities. All but one were Muslim, which explains the need for sanitation water used for cleansing. Shared boreholes and wells were the predominant water sources (Table 5).

Most respondents expressed greatest concern about mosquitoes that bite at night and cause malaria. The respondent's risk category did not impact household mosquito management or disease prevention behaviors. Several women and children distinguished between mosquitoes that bite during the day and those that bite at night; mentioning that day-biting mosquitoes are present but harmless. One woman stated that mosquitoes do not bite during the day at all. On the other hand, two women stated that mosquitoes only affect them during the day (Table 6).

All of the women and children interviewed stated that mosquitoes primarily cause malaria and that at least one person in each family had been severely affected by malaria. Several respondents described other mosquito-borne diseases accurately (filariasis, chikungunya, and bilharzia), while others stated that non-mosquito-borne diseases are caused by mosquito bites (e.g., scabies, typhoid, cholera, and pneumonia) (Table 6).

Respondents demonstrated limited knowledge about the mosquito life cycle. Few mentioned household containers as primary breeding habitats because they are considered relatively clean. One woman described how mosquitoes preferentially breed in dirty water in coconut shells, saying: *"[The water] stays today, tomorrow, and the third day is when they [mosquitoes] get in there. . .. They normally wait until they get some bad smell from the water inside the shell"* (Table 6).

When researchers showed a video depicting larvae and pupae, less than one quarter of respondents recognized them as immature mosquitoes. Most considered the larvae and pupae to be a type of "unclean" organism, such as bacteria, parasites, or worms that could cause stomach infections and diarrheal disease. They did not recognize that the larvae and pupae would transform into a flying adult mosquito. Local names for the immature mosquitoes ranged widely: *mwamtibwiri, mwamchibwiri,* and *vitikutiku* (Kidigo words describing the wriggling movement); *vimelea* (algae); *sungusungu* (ants); *jiggers* (chigoe fleas); and *vidudu* (a Kiswahili term for bacteria or bugs), or maggots. (Table 6)

When researchers asked if respondents would do anything if they saw immature mosquitoes in their water, more women than children described the importance of pouring out the water or at least not drinking the larvae or pupae. One woman emphasized this point stating, *"You see them [immature mosquitoes] do this [using her finger to demonstrate the wriggling*

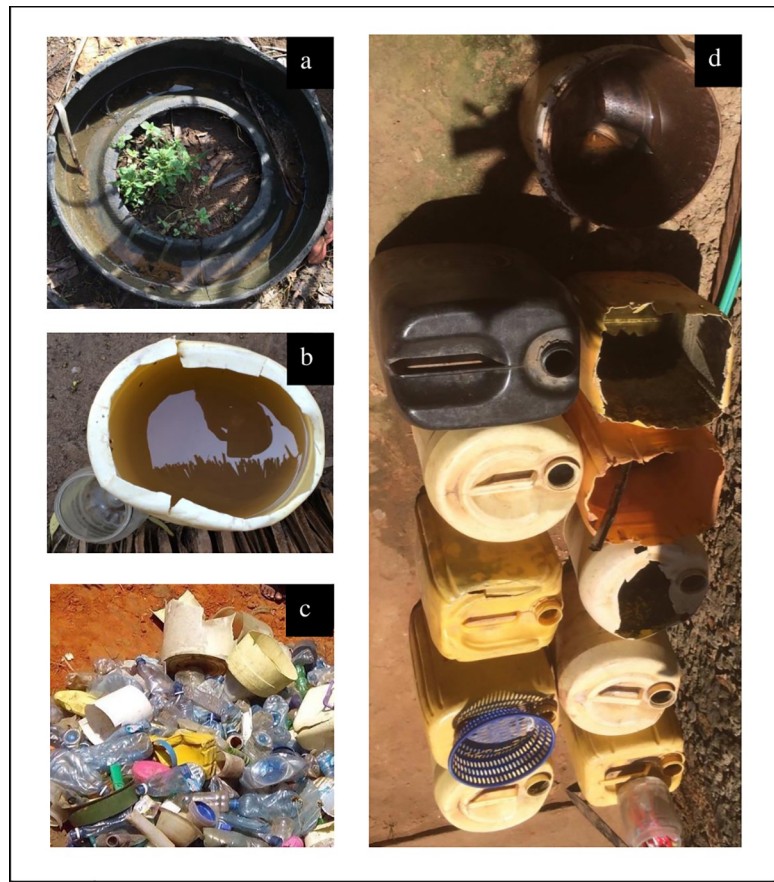

**Fig 1.** Examples of productive mosquito habitats: a) tires with no immediate purpose, b) bucket and small container for sanitation, c) small domestic containers with no immediate purpose, and d) buckets and jerrycans for laundry or with no immediate purpose.

**Table 1. Table of mosquito habitats by type and purpose among 444 entomological surveys in Kwale County, Kenya, between September-December 2016.** Percentage of total habitats are shown in parentheses across type and purpose categories. Percent of total immature mosquitoes (both larvae (early and late instars) and pupae) are reported within the cells of the table with shaded color highlighting with green, yellow, orange, and red representing 0%, 0–5%, 5–20%, and >20% of larval abundance, respectively. Habitat type according to size: 1) small domestic containers, vases, and cooking vessels (<5L), 2) tires, buckets, jerrycans, and basins (10-25L), and 3) drums and tanks (>25L).

| | | Habitat type (% of habitats) | | | | | | | |
|---|---|---|---|---|---|---|---|---|---|
| | | Bucket (48.1) | Tire (0.7) | Small containers (9.2)[1] | Basin (6.5) | Drum (2.9) | Jerrycan (28.8) | Other (3.8)[2] | Total |
| Purpose (% habitats) | No immediate purpose (3.4) | 13.8 | 28.0 | 13.4 | 0.0 | 0.0 | 0.0 | 0.0 | 55.2 |
| | Laundry (34.4) | 37.4 | 0.0 | 0.0 | 1.0 | 0.0 | 2.9 | 0.0 | 41.3 |
| | Sanitation (12.5) | 1.9 | 0.0 | 0.1 | 0.1 | 0.0 | 0.0 | 0.0 | 2.2 |
| | Animals (2.3) | 0.0 | 0.0 | 0.7 | 0.0 | 0.0 | 0.0 | 0.0 | 0.7 |
| | Plants (0.3) | 0.0 | 0.0 | 0.4 | 0.0 | 0.0 | 0.0 | 0.0 | 0.4 |
| | Other (47.1)[3] | 0.0 | 0.0 | 0.0 | 0.0 | 0.2 | 0.0 | 0.0 | 0.2 |
| | Total | 53.1 | 28.0 | 14.7 | 1.2 | 0.2 | 2.9 | 0.0 | 100.0 |

[1]Food containers, bottles, vases

[2]Tanks and cooking vessels

[3]Bathing, drinking, cooking, and multiple functions

**Table 2. Larval and pupal productivity profiles based on habitat type.**

| Habitat type | No. of containers | No. of containers filled with water | % filled with rainwater | No. of positive containers | % of positive containers filled with rainwater | No. of early instars | No. of late instars | No. of pupae |
|---|---|---|---|---|---|---|---|---|
| Buckets | 1096 | 860 | 16.3 | 12 | 75.0 | 206 | 348 | 225 |
| Jerrycan | 683 | 514 | 15.8 | 4 | 100.0 | 6 | 18 | 18 |
| Small containers | 320 | 164 | 23.2 | 8 | 62.5 | 122 | 47 | 46 |
| Basin | 174 | 116 | 22.4 | 3 | 66.7 | 0 | 6 | 11 |
| Drum | 56 | 52 | 38.5 | 1 | 0.0 | 0 | 3 | 0 |
| Tire | 29 | 12 | 91.7 | 6 | 100.0 | 102 | 219 | 90 |
| Other | 93 | 68 | 35.3 | 0 | 0.0 | 0 | 0 | 0 |

movement]; then you won't drink that water. You just take the water and pour it down, because a young kid will just get the water without knowing that those things can cause disease. Or if you have dirty clothes, then you just use the water to clean them."

One respondent described the importance of not knowingly ingesting the contaminated water even though it happens accidentally: *"I just clean it and pour the water, but if you haven't seen them because it's at night you'll just drink them, just only one of it. . . you get a stomach ache"*. Then when asked why she wouldn't fetch water more frequently to avoid "bugs" from entering her water, she said, *"Where is the time to fetch water*?! *I want to go to the shamba (field for farming). . . I'm tired."* Several women echoed this sentiment and drew the connection between water scarcity and mosquito breeding. They specifically bemoaned the lack of piped-water access, which necessitates storing water for long periods of time and results in wriggling worms, bacteria, and generally unclean water (Table 6).

Women and children mostly reported mosquito avoidance measures, such as sleeping under bednets, as the most effective way to minimize mosquito-borne diseases. About one-third of respondents reported learning about bednets from doctors while being treated for malaria and other diseases at hospitals and school-based community clinics. Three women mentioned that they learned by experience, witnessing cause and effect. If they lit a fire, for example, they noticed how mosquitoes fled and they did not get bitten.

Few respondents knew about, let alone practiced, source reduction as a way to prevent *Ae. aegypti* mosquito breeding. Covering water, for example, was a measure that women took to avoid contaminating water for drinking and cooking, but was not an intentional source reduction action.

## Discussion

This study combines entomological surveys that identify the most productive mosquito breeding habitats with qualitative interviews that explore behaviors related to source reduction. Our

**Table 3. Larval and pupal productivity profiles based on water source (among the 1,786 containers filled with water).**

| Water source | No. of containers | No. of positive containers | No. of early instars | No. of late instars | No. of pupae |
|---|---|---|---|---|---|
| Rain | 340 | 25 | 429 | 598 | 379 |
| Borehole | 461 | 4 | 0 | 22 | 7 |
| Well | 581 | 4 | 3 | 21 | 4 |
| Tap | 393 | 1 | 4 | 0 | 0 |
| Dam | 11 | 0 | 0 | 0 | 0 |
| River/stream | 0 | 0 | 0 | 0 | 0 |

**Table 4. Larval and pupal productivity profiles based on container purpose (among the 1,786 containers filled with water).**

| Container purpose | No. of containers | No. of positive containers | No. of early instars | No. of late instars | No. of pupae |
|---|---|---|---|---|---|
| Bathing | 180 | 0 | 0 | 0 | 0 |
| Drinking | 92 | 1 | 0 | 3 | 0 |
| Cooking | 77 | 0 | 0 | 0 | 0 |
| Animals | 41 | 1 | 0 | 10 | 0 |
| Plants | 5 | 1 | 0 | 5 | 1 |
| No immediate purpose | 60 | 14 | 268 | 400 | 142 |
| Laundry | 614 | 12 | 161 | 202 | 243 |
| Sanitation | 225 | 5 | 7 | 21 | 4 |
| Other/multiple functions | 497 | 0 | 0 | 0 | 0 |

results provide a deeper understanding of the social ecological context and allow us to recommend vector control strategies. The combination of low entomological infestation, low perceived risk of daytime mosquitoes, and limited awareness about mosquito breeding in manmade containers, suggests that interventions in this part of coastal Kenya should be targeted so as to require minimal effort and align with existing incentives.

**Table 5. Demographic characteristics of 40 female caregivers participating in in-depth interviews and structured observations in Kwale County, Kenya, July-August 2016.**

| Characteristic | Frequency (%) |
|---|---|
| Age (years)[1] | 37.2 (9.2) |
| Education (years)[1] | 4.7 (3.6) |
| Religion | |
| Muslim | 39 (97.5) |
| Christian | 1 (2.5) |
| Marital status | |
| Unmarried | 1 (2.5) |
| Married | 35 (87.5) |
| Divorced/separated | 1 (2.5) |
| Widowed | 3 (7.5) |
| Occupation | |
| Farmer | 24 (60.0) |
| Business owner | 7 (17.5) |
| Teacher | 1 (2.5) |
| House help | 1 (2.5) |
| Housewife | 7 (17.5) |
| Children (number)[1] | 2.2 (1.9) |
| Household residence (years)[1] | 14.9 (9.7) |
| Water source | |
| Borehole | 19 (47.5) |
| Well | 15 (37.5) |
| Public tap | 3 (7.5) |
| River | 1 (2.5) |
| Dam | 1 (2.5) |

[1]Mean (standard deviation) reported for continuous variables.

**Table 6. Coding themes and illustrative quotes from semi-structured in-depth interviews with 35 female caregivers and 37 children in Kwale County, Kenya, July-August 2016.**

| Theme | Sub-theme | Female caregiver frequency (%)[1] | Child frequency (%)[2] | Quotes and examples |
|---|---|---|---|---|
| Risk perception of mosquito types | Night-timing biting mosquitoes affect us most | 30 (86) | 30 (81) | "The night mosquito is the one that hurts, the night one. Egheee, there are those that I have told you they come at 2 or 1 at night, and then there are the daytime ones. Mmmmmh, the night one is the one which disturbs, always ndyeeee [making the noise that mosquitoes make when they fly] but the daytime one doesn't disturb." (V02001) "They bite more at night especially starting from midnight. That's when the dangerous mosquitoes bite and cause malaria." (V01005) "Mosquitoes also bite during the day but those that bite during the day are not harmful at all. . . harmful mosquitoes are available at night." (V01040) |
| | Only night-time biting mosquitoes (not day-time) cause harm | 4 (11) | N/A | |
| Mosquito-borne diseases | Malaria | 35 (100) | 37 (100) | |
| | Filariasis | 4 (11) | 1 (3) | |
| | Chikungunya | 3 (9) | 0 (0) | |
| | Bilharzia | 0 (0) | 3 (8) | |
| | Scabies | 2 (6) | 0 (0) | |
| | Cholera | 0 (0) | 1 (3) | |
| | Typhoid | 2 (6) | 0 (0) | |
| | Pneumonia | 2 (6) | 0 (0) | |
| | Others | 4 (11) | 4 (11) | Umbilical cord enlargement, headache, dizziness, diarrhea, stomach or blood vessel disease |
| Knowledge of larvae and pupae | Appear in _____ weeks. . . | | | |
| | Unknown | 13 (37) | 8 (22) | |
| | < 1 | 16 (46) | 14 (38) | |
| | 1–2 | 5 (14) | 5 (14) | |
| | >2 | 1 (3) | 1 (3) | |
| | Are young mosquitoes | 8 (23) | 6 (19) | "It's only you. . . when you came and you sieved them and called them mosquitoes. . . [now] I also call them mosquitoes." (V02045) "[I] thought they were water insects but some KEMRI researchers came and told me they were mosquitoes." (V01014) |
| | Have a negative effect if ingested | 29 (83) | 22 (59) | |
| | Cause stomach issues | 19 (54) | 11 (30) | "You'll just feel it in your stomach, if you want a disease then drink them." (V02003) "They scare me when I see them. They have effects. . . for example when they go inside someone's stomach. It is diseases." (V01036) Descriptions ranged from general stomach infections to diarrhea, or parasitic worms like bilharzia. |
| | Cause mosquito problems | 5 (14) | 3 (8) | Some mentioned that swallowing the larvae and pupae caused malaria while others mentioned that they would become adult mosquitoes. |
| | If found in my water I will. . . | | | "As for me, I have a well nearby. I'll just pour the water down and then fetch some new ones. But for those who get their water from a distance because also me before we dug that well I used to get my water very far. So it's not good to pour the water. I used to sieve the water. Or when there is no sieve I take a clean cloth and use it to sieve the water from the insects." (V01014) |
| | Pour my water out | 19 (54) | 9 (24) | |
| | Only use the water for washing/bathing but never drinking or cooking | 8 (23) | 3 (8) | |
| | Drink the water as normal/do nothing | 1 (3) | 7 (19) | |
| | Treat the water or otherwise remove them | 7 (20) | 4 (11) | |

*(Continued)*

**Table 6.** (Continued)

| Theme | Sub-theme | Female caregiver frequency (%)[1] | Child frequency (%)[2] | Quotes and examples |
|---|---|---|---|---|
| Protective behaviors | Sleeping under bed nets | 31 (89) | 32 (86) | "The nets don't cover the beds properly, the nets can only cover a school or a hospital bed, but if it's a family size bed where by two to three children sleep together. . . the nets are small." (V01014) |
| | Deterring or killing adult mosquitoes | 10 (29) | 12 (32) | Mosquito coils, fire, killing mosquitoes or wearing long sleeve shirts and pants. |
| | Cleaning the environment | 17 (49) | 23 (62) | ". . .. if you are sleeping under nets and yet your environment is not clean, that won't help." (V01012) Sweeping and clearing bushes or grasses. |
| | Source reduction | 10 (29) | 6 (16) | Collecting/burning coconut shells, covering containers, turning containers upside down or removing stagnant water. |

[1]Thirty-five female caregivers responded

[2]Thirty-seven children responded

To the extent possible, vector control strategies should aim to identify and target high-risk households. In the study region, *Ae. aegypti* larvae and pupae were over-dispersed, meaning that immature mosquitoes were concentrated in relatively few containers at a small number of households [17]. This highlights the potential utility of tools like the Premises Condition Index (PCI) that have been developed from predictive models to identify high-risk households for targeted vector control [18, 19]. The PCI, originally developed in Australia and further tested in Central America and South Asia, aims to rapidly assess the cleanliness of an area and the degree of shade in order to predict the risk of *Ae. aegypti* infestation [19–21]. This or a similar tool could be further honed and adapted for the study region and other Sub-Saharan African countries.

Because only a few habitat types predominated, source reduction should target highly productive habitat types. Many interventions encourage targeted source reduction based on container type (e.g., buckets, drums, tanks, and tires,) but fewer consider container purpose [22]. From a behavioral standpoint, considering purpose when targeting productive habitat types would reduce the number of containers of concern dramatically and would also increase the impact of any efforts [23]. The most productive habitat types can be grouped into three categories based on purpose: 1) containers with an immediate purpose or with a potential future purpose (e.g., buckets), 2) containers with no immediate purpose but with repurposing value (e.g., tires) and 2) containers with no immediate purpose and limited repurposing value (e.g., small domestic containers and bottles).

Buckets used for laundry or those kept around for some future use could be covered if covers were easy to retain. Covering containers is a commonly recommended source reduction tactic and has been found to significantly reduce the odds of a container having immature mosquitoes by more than 80% [17]. Given the irregularity of laundry bucket use, covering with nylon net could allow for the continued use of containers without removing the cover. Nylon net covers have reduced mosquito abundance elsewhere, despite some long-term maintenance needed to patch any holes that the form in the net [24]. In this context, residents would need to be convinced that it would be worth their time to cover buckets used for purposes other than drinking and cooking since respondents did not see a reason to cover water that wasn't being ingested. One challenge with covering buckets is related to the number of buckets in circulation. Although targeting buckets would reduce mosquito breeding by half in

this study region, with more than 1,000 buckets, it would be time intensive to manage and sustain.

Tires, on the other hand, were highly productive and yet few in number, making them an attractive source reduction target. Consistent with the evidence from this study, tires have been found to be highly productive habitats elsewhere in Kenya and other countries in sub-Saharan Africa [8, 25–27], as well as across the world, in the US, Caribbean, South and Southeast Asia [20, 28–30]. Part of the reason tires may be so productive is due to the fact that they sit for long periods of time undisturbed, as we noted in this study. Other factors could include the water temperature and detritus that tends to collect in tires, making them ideal breeding sites for numerous *Aedes* and *Culex* species [20].

Tires in this study context had no immediate purpose but considerable value. Since covers are unlikely to be applicable to tires that are not intended to hold water, we recommend different actions. Some tires remain outside residences because they are informally used as seats. Cutting and turning over these tires could ensure that they don't collect water. Others could be collected and re-purposed to make recycled goods such as toys or shoes.

For small domestic containers, food tins, and plastic bottles with no purpose at all, we recommend community clean-ups and efforts to improve solid waste management. In the short-term, households could consolidate trash under a shaded storage place away from rain. Periodic community-led trash clean-ups may be more appropriate than household-level actions since they would not require a change in habits. Since the respondents already expressed interest in maintaining their compounds, any additional benefit or income that they could generate from collecting, recycling, or re-using no-purpose containers would add even more incentive. At a larger scale, improving centralized solid waste management and access to piped water would have benefits for long-term vector control as well as the prevention of other diseases [31]. However, governmental provision and maintenance of these services will take time, and coverage is likely to be patchy and inconsistent, especially in informal and rural settlements [32].

Our source reduction recommendations are specific to the study region. Given the low mosquito infestation indices, our data suggest that *Ae. aegypti*-specific control measures like targeted source reduction may be easy to implement but a lower priority than *Anopheles* control. By only sampling outdoors, we may have underestimated the abundance of *Ae. aegypti* immatures. However, the effect of this is likely to be minimal given the evidence that *Ae. aegypti* primarily breed in outdoor containers across the study region [8].

Future research should consider urban areas of coastal Kenya where *Ae. aegypti* mosquitoes have been found to be three times more abundant than nearby rural areas [8, 9]. Numerous outbreaks of chikungunya have occurred in cities along Kenya's coast within the past decade [33, 34]. Although disease risk is likely to be higher in urban areas, these communities tend to be more informal and less cohesive. Therefore, source reduction recommendations may benefit less from collaborative community clean-ups and necessitate vector control strategies tailored to the unique social and ecological characteristics of those urban settings.

## Supporting information

**S1 Text. Semi-structured interview guide (female caregiver).**
(DOCX)

**S2 Text. Semi-structured interview guide (school child).**
(DOCX)

**S3 Text. Semi-structured interview guide (female caregiver)–Swahili version.**
(DOCX)

**S4 Text. Semi-structured interview guide (school child)–Swahili version.**
(DOCX)

**S1 Table. Table of mosquito habitats by type and purpose among 444 entomological surveys in Kwale County, Kenya, between September-December 2016.** Percentage of total habitats are shown in parentheses across type and purpose categories. Percent of total larvae (early and late instars) are reported within the cells of the table with shaded color highlighting with green, yellow, orange, and red representing increasing abundance.
(DOCX)

**S2 Table. Table of mosquito habitats by type and purpose among 444 entomological surveys in Kwale County, Kenya, between September-December 2016.** Percentage of total habitats are shown in parentheses across type and purpose categories. Percent of total pupae are reported within the cells of the table with shaded color highlighting with green, yellow, orange, and red representing increasing pupal abundance.
(DOCX)

## Acknowledgments

The authors appreciate remarkable field data collection support from Julius Kamoni, Robin Bundi, Hussein Kitsongo, Rashid Mtowa. Finally, the authors acknowledge the study participants in Kwale County.

## Author Contributions

**Conceptualization:** Jenna E. Forsyth, Francis M. Mutuku, Lydiah Kibe, A. Desiree LaBeaud.

**Data curation:** Jenna E. Forsyth, Francis M. Mutuku, Chika Egemba.

**Formal analysis:** Jenna E. Forsyth, Francis M. Mutuku, Chika Egemba.

**Funding acquisition:** Jenna E. Forsyth, Francis M. Mutuku, A. Desiree LaBeaud.

**Methodology:** Jenna E. Forsyth, Francis M. Mutuku, Lydiah Kibe, Luti Mwashee, Joyce Bongo, Nicole M. Ardoin, A. Desiree LaBeaud.

**Project administration:** Francis M. Mutuku, A. Desiree LaBeaud.

**Supervision:** Francis M. Mutuku, A. Desiree LaBeaud.

**Visualization:** Jenna E. Forsyth.

**Writing – original draft:** Jenna E. Forsyth.

**Writing – review & editing:** Jenna E. Forsyth, Francis M. Mutuku, Lydiah Kibe, Nicole M. Ardoin, A. Desiree LaBeaud.

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
