## [Decision Letter · Decision Letter 0]

14 Jan 2020

Dear Dr. Forsyth:

Thank you very much for submitting your manuscript "Source reduction with a purpose: mosquito ecology and community perspectives offer insights for improving household mosquito management in coastal Kenya" (#PNTD-D-19-01882) for review by PLOS Neglected Tropical Diseases. Your manuscript was fully evaluated at the editorial level and by independent peer reviewers. The reviewers appreciated the attention to an important problem, but raised some substantial concerns about the manuscript as it currently stands. These issues must be addressed before we would be willing to consider a revised version of your study. We cannot, of course, promise publication at that time.

We therefore ask you to modify the manuscript according to the review recommendations before we can consider your manuscript for acceptance. Your revisions should address the specific points made by each reviewer. 

When you are ready to resubmit, please be prepared to upload the following:

(1) A letter containing a detailed list of your responses to the review comments and a description of the changes you have made in the manuscript.

(2) Two versions of the manuscript: one with either highlights or tracked changes denoting where the text has been changed (uploaded as a "Revised Article with Changes Highlighted" file); the other a clean version (uploaded as the article file).

(3) If available, a striking still image (a new image if one is available or an existing one from within your manuscript). If your manuscript is accepted for publication, this image may be featured on our website. Images should ideally be high resolution, eye-catching, single panel images; where one is available, please use 'add file' at the time of resubmission and select 'striking image' as the file type. 

Please provide a short caption, including credits, uploaded as a separate "Other" file. If your image is from someone other than yourself, please ensure that the artist has read and agreed to the terms and conditions of the Creative Commons Attribution License at http://journals.plos.org/plosntds/s/content-license (NOTE: we cannot publish copyrighted images). 

(4) If applicable, we encourage you to add a list of accession numbers/ID numbers for genes and proteins mentioned in the text (these should be listed as a paragraph at the end of the manuscript). You can supply accession numbers for any database, so long as the database is publicly accessible and stable. Examples include LocusLink and SwissProt.

(5) To enhance the reproducibility of your results, we recommend that you deposit your laboratory protocols in protocols.io, where a protocol can be assigned its own identifier (DOI) such that it can be cited independently in the future. For instructions see http://journals.plos.org/plosntds/s/submission-guidelines#loc-methods

While revising your submission, please upload your figure files to the Preflight Analysis and Conversion Engine (PACE) digital diagnostic tool, https://pacev2.apexcovantage.com/ PACE helps ensure that figures meet PLOS requirements. To use PACE, you must first register as a user. Then, login and navigate to the UPLOAD tab, where you will find detailed instructions on how to use the tool. If you encounter any issues or have any questions when using PACE, please email us at figures@plos.org.

We hope to receive your revised manuscript by Mar 14 2020 11:59PM. If you anticipate any delay in its return, we ask that you let us know the expected resubmission date by replying to this email.

To submit a revision, go to https://www.editorialmanager.com/pntd/ and log in as an Author. You will see a menu item call Submission Needing Revision. You will find your submission record there. 

Sincerely,

Roberto Barrera, Ph.D.

Associate Editor

Eric Dumonteil

Deputy Editor

Reviewer's Responses to Questions

**Key Review Criteria Required for Acceptance?**

**Methods**

-Are the objectives of the study clearly articulated with a clear testable hypothesis stated?

-Is the study design appropriate to address the stated objectives?

-Is the population clearly described and appropriate for the hypothesis being tested?

-Is the sample size sufficient to ensure adequate power to address the hypothesis being tested?

-Were correct statistical analysis used to support conclusions?

-Are there concerns about ethical or regulatory requirements being met?

Reviewer #1: This is an interesting study, backed with clear objectives and well designed to address the insights for improving household mosquito management in coastal Kenya. There is a clear statement on the sample size used for entomological and interviews components. 

HOWEVER, 

a) In the abstract- Looking at Methodology/ Principal Methodology: these two items could be more clear if separated!

b) Lines - 33-34: “The field team conducted entomological surveys in 34 444 households in Kwale County, coastal Kenya, between May and December 2016.” The abstract does not include the interviews— was that deliberately or an oversight? 

Main Document

Methodology

Sampling approach applied to obtain households both for entomological and Interviews need to be presented in a more straightway style. 

Ref. Lines 152-154: While it is clearly written that “Fifty houses with children in 152 grades 5 and 6 (approximately ages 11 to 16) were randomly selected from 10 different primary school rosters” . Readers at Ref Lines No. 168-172: will question “How were these 40 households selected?”

Reviewer #2: The authors are investigating the productive mosquito breeding habitats and explore the existing knowledge, attitudes and behaviour of the community participants in relation to source reduction. They used mixed methods approach that combines entomological surveys and interviews to female caregivers and children. This approach proves better as it provides a deeper understanding of the social ecological context towards vector control strategies. This is a well written manuscript which I think should be published without delay. 

Introduction/Background: The background information is good, well written and the research question is easily identifiable.

Materials & Methods: The methodologies applied are appropriate and adequately described. The study design is appropriate and the data analysis is also adequately described.

Reviewer #3: A major method (how purpose of containers was defined and studied) needs to be better described.

The population wasn't necessarily the best choice given the low infestation indices found, but this did not affect my recommendation.

My recommendation is based on the need to describe that method better as well as make correct use of the numerous references, and shorten and sharpen the results and discussion.

All other aspects of study design are appropriate.

**Results**

-Does the analysis presented match the analysis plan?

-Are the results clearly and completely presented?

-Are the figures (Tables, Images) of sufficient quality for clarity?

Reviewer #1: The manuscript contains quantitative and qualitative results. However, the Authors there is no description on Data analysis. Hence, The authors are expected to address this gap!

Reviewer #2: Results, Discussion and conclusion:The results and discussions are coherently described. The Tables and supporting information provided are of sufficient quality and well presented. The discussion highlights the different aspects of the findings and are neatly discussed in relation to the current literature.

Reviewer #3: The analysis presented matches the analysis plan.

The results are presented in a repetitive way at times. I suggest in my comments how to shorten.

The figures are of sufficient quality for clarity and don't require that much more text as the volume currently provided.

**Conclusions**

-Are the conclusions supported by the data presented?

-Are the limitations of analysis clearly described?

-Do the authors discuss how these data can be helpful to advance our understanding of the topic under study?

-Is public health relevance addressed?

Reviewer #1: Conclusions: This is missing in the main document while abstract prepares readers to see it!

Reviewer #2: The conclusions are clear and logical and are supported by data presented.

Reviewer #3: The conclusions are supported by the data presented, but they are currently presented in a vague manner.

There is no limitations section and I recommend one in my comments.

All other aspects of conclusions are appropriate.

**Editorial and Data Presentation Modifications?**

Reviewer #1: "Minor Revision"

Reviewer #2: N/A

Reviewer #3: 10. Various other comments throughout:

1. Water “holding” might be a better term than water “storage” containers, to better capture the fact that in this setting the water collection in the containers without immediate purpose was not intentional.

2. Lines 73-75. Consider this editing: “Individuals with these diseases can range from asymptomatic to… suffering from life-threatening encephalitis and hemorrhage, to… debilitating arthritis that can persist for years (Murray et al. 2013). 

3. Line 87. “Adult mosquitoes typically venture less than 100 meters from their hatch site” – true if they hatch in and around people’s homes because they do not need to travel far for a blood meal. Not true if they hatch at a tire dump or car graveyard with few or no humans around. Consider qualifying statement.

4. Line 88 re biting during circumscribed times of the day – qualify that this finding pertains to Kenya.

5. Lines 90-93. Statement is true about any setting.

6. Line 98 “easier” tasks vs. easy tasks because covering containers is not easy if you actually try it…

7. Lines 117-118. “Most interventions do not target specific habitat types, possibly because mosquito habitats are numerous and vary by season.” All the interventions that you cite target specific habitat types. The point you want to make, not here – in the discussion, is that interventions should target containers by purpose. 

8. Line 126. If you only looked at containers outside homes without checking inside homes, you cannot claim that the important containers were only found outside. In fact this point will fit well in a study limitation section.

9. Line 143. “Mosquitoes are least abundant during the long dry season.” This may be true for Anopheles. Is there any evidence for Aedes? Better off to delete. Or qualify which mosquitoes you refer to.

10. Lines 211-214. Specify if the containers found had water or not. If they were dry, it was no surprise that they didn’t have immature mosquitoes.

11. Line 404. “For example, instead of non-targeted source reduction, requiring households to consider 2,452 containers…” A reason to target dry but useful containers would be to protect them from the rain. Rephrase.

13. Lines 409-411: “While there were only 29 tires, these accounted for up to 30% of immature mosquitoes, so targeting tires would be less behaviorally intensive than buckets, but not as effective as considering purpose along with habitat type.” You didn’t find any tires with a purpose (Table 1) so it would be just as effective to target those 29 tires. A more important problem is that tires are kept for future use, and those without any purpose at all have nowhere to go - no services exist...

**Summary and General Comments**

Reviewer #1: This manuscript is generally well written, with relevant materials in its field of study. It is publishable upon addressing the minor revisions as suggested.

Reviewer #2: The paper is well written and appropriate for the journal

Reviewer #3: The manuscript incudes an excellent title and well written abstract, both of which raise the reader’s expectations for a very good piece of research with important intervention insights. However some key but fixable weaknesses need to be addressed before the manuscript can bring its point across to the reader.

1. Purpose takes a central focus on the title, abstract and throughout the manuscript. However, there is no information on how purpose was identified. This is a major weakness of the manuscript. In the methods section, please, provide explicit information on how the purpose of each container studied was defined. Include who reported the purpose, and how was the question asked (open ended/closed). The meaning of “no purpose” may well be “no immediate purpose, but we keep the container for future use.” Indeed, it takes the reader until page 26 of the manuscript to find out that indeed this was the case for the larger containers. The cut tire e.g. in Fig 1a may not have had a purpose at the moment of the survey, but the way it is cut is characteristic of tires used as animal drinking dishes. This reviewer therefore suggests that “no purpose” is replaced with “no immediate purpose” throughout the manuscript and tables, and to explain this in the methods and take it out of page 26.

2. The current use of citations in the text, especially in the introduction raises serious concerns. The manuscript should not be accepted before the authors go over each citation and check that it is used correctly to make the point that it is intended to make. This is another major weakness of the manuscript. Examples include:

1. Gubler 2002 does not mention child development. Rephrase

2. Lines 84-86. Harrington 2005 does not seem an appropriate reference because it talks about release and recapture of adult mosquitoes in Puerto Rico and Thailand – not where they breed.

3. Ngugi found key containers outdoors in the western region of Kenya and indoors in the coastal region. Drums, buckets and pots that Ngugi found were purposeful water storage containers. It should not be taken for granted that the important containers are found only outdoors. This may vary by setting. A similarly inaccurate statement about Ngugi is made in lines 119-121.

4. Lines 95-97, re Keating reference: when the goal is to prevent malaria, bed nets will be effective but source reduction of containers around the home will not be effective, because Anopheles doesn’t breed in containers around the home like Aedes. So this reference here doesn’t apply and malaria control programs should not bring reduction of water containers in the home into the mix! Same goes for Kibe reference – larvae found in and around homes are not anopheles larvae and controlling them will not control malaria. The point to make here instead is that the overwhelming focus has been in malaria control and the Aedes larvae found in and around the home are allowed to complete their life cycle undetected. But this means that education efforts should distinguish between mosquito larvae and adults and specifically mention which behavior will control which type of mosquito larvae.

5. Lines 100-102. Neither Andersson nor Winch say that households often perceive source reduction labor intensive or ineffective. In fact in Andersson, households had a positive attitude toward source reduction. Andersson et al perceive that “We do not, however, expect community participation in dengue control [via source reduction] to be easy or easily sustainable.” Winch et al perceive that source reduction behaviors are complex in nature, large in number and difficult to perform by households; that entomologists are concerned that voluntary source reduction may be ineffective because of the high level of coverage required. Rephrase or delete.

6. Line 103. The intervention by Arunachalam 2012 was targeted but definitely not simple - covers had to be made in three different sizes and promoted through community actors, clean-up campaigns had to be organized, and dissemination of dengue information given through schoolchildren. The point to make here is that good covering interventions exist and could be leveraged for your study site. The point with its reference is better suited for the discussion. 

7. Lines 104-107. In Bowman 2016, “house screening significantly reduced dengue risk, OR 0.22 (95% CI 0.05–0.93, p = 0.04), as did combining community-based environmental management and water container covers, OR 0.22 (95% CI 0.15–0.32, p<0.0001).” Refrain from singling out covers. 

8. Lines 121-124. Prochaska 2008, in contrast to what the authors state, provides a rationale and need for multiple health behavior change interventions, and reviews a series of theoretical and methodological issues that need to be resolved in order to implement such multiple health behavior changes interventions successfully. In addition all the successful source reduction interventions that the authors cite earlier consist of implementing multiple health behaviors at a time. Rephrase. 

9. Lines 364-367: “The implication of over-dispersed larval and pupal distributions on vector control is that an intervention must obtain good geographic coverage in order to reduce breeding enough to disrupt disease transmission (Irvine et al. 2018).” This statement doesn’t make sense in the context of overdispersal. Overdispersal calls for finding the heavily infested households and concentrating on those. Besides, the Irvine reference is on a totally different mosquito with very different larval habitats not studied in your paper. Irvine studies biting behavior instead. The statement can go.

10. Line 376. “Consistent with other studies, tires, buckets and small domestic containers were the most productive habitat types.” In the Focks and Chadee 1997 study cited to defend this point, tires, buckets and small domestic containers were equally important as water storage drums and tanks. In that setting (Trinidad) water storage drums and tanks are much larger in volume (55 US gallons, 1 US gallon equals 3.78 litres). Therefore this is not a good comparison. And the Ndenga and Ngugi studies also found important indoor containers. You are better off eliminating that point altogether.

11. Kathomi 2013. Complete reference or delete

3. In the abstract, methods and other places, knowledge and attitudes are mentioned as behavioral determinants studied in the manuscript. However, the reviewer could not identify any data pointing to attitudes. Knowledge, yes, as well as a number of other determinants are described, such as people’s previous lived experience with mosquitoes (felt night biting, noticed night mosquito nuisance, felt effectiveness of bednets); the purpose of the water held in the containers is a major behavioral determinant in the study; lack of access to adequate covers for their containers, and lack of access to waste collection services are mentioned in the discussion. This reviewer therefore suggests replacing knowledge, attitudes and behaviors with, say, exploration of household mosquito management behaviors and their behavioral determinants, or similar. 

4. The study is described as combining mixed methods. This reviewer suggests stating combining entomological surveys with qualitative methods, wherever mixed methods are currently mentioned. 

5. The results section text is often repetitive, as well as repeats what is stated in the tables. Please go through and shorten. Examples include: Lines 247-269, lines 277-279, Text repeating much of Table 6. 

6. A glaring finding of the study is the low entomological infestation indices, making any intervention easier to implement, especially due to the documented overdispersal that calls for targeting certain households rather than everyone, but at the same time not a highly prioritized site for Aedes aegypti control. The manuscript will benefit from commenting on this in the discussion, or in a separate study limitation section. 

7. In abstract, author summary and discussion, qualify conclusions as appropriate for the specific study setting, therefore recommendations applying to that setting, not necessarily generally, because in many studies cited in the manuscript, the purposeful containers are more problematic than or as problematic as the accidental ones.

8. Please shorten and sharpen the discussion. No need to repeat results. Limit and sharpen recommendations to the specific study site; e.g. instead of mentioning covering in general, state what type of covers would serve buckets of various sizes. Limit each recommendation to one or two sentences. The reader is exhausted by the time we reach the discussion!

9. In line with an earlier comment, separate out the household mosquito management recommendations in the discussion, for containers with no purpose at all vs. those with no immediate purpose, from the start.

PLOS authors have the option to publish the peer review history of their article (what does this mean?). If published, this will include your full peer review and any attached files.

Reviewer #1: Yes: Adiel K. Mushi

Reviewer #2: No

Reviewer #3: No

---

## [Editor Report · Decision Letter 1]

20 Mar 2020

Dear Dr. Forsyth,

We are pleased to inform you that your manuscript 'Source reduction with a purpose: mosquito ecology and community perspectives offer insights for improving household mosquito management in coastal Kenya' has been provisionally accepted for publication in PLOS Neglected Tropical Diseases.

Best regards,

Roberto Barrera, Ph.D.

Associate Editor

Eric Dumonteil

Deputy Editor

---

## [Editor Report · Acceptance letter]

4 May 2020

Dear Dr. Forsyth,

We are delighted to inform you that your manuscript, "Source reduction with a purpose: mosquito ecology and community perspectives offer insights for improving household mosquito management in coastal Kenya," has been formally accepted for publication in PLOS Neglected Tropical Diseases.

Best regards,

Serap Aksoy

Editor-in-Chief

Shaden Kamhawi

Editor-in-Chief
